# Long COVID in cancer patients: preponderance of symptoms in majority of patients over long time period

Hiba Dagher[1], Anne-Marie Chaftari[1]\*, Ishwaria M Subbiah[2], Alexandre E Malek[1], Ying Jiang[1], Peter Lamie[3], Bruno Granwehr[1], Teny John[1], Eduardo Yepez[1], Jovan Borjan[4], Cielito Reyes-Gibby[5], Mary Flores[5], Fareed Khawaja[1], Mala Pande[6], Noman Ali[3], Raniv Rojo[7], Daniel D Karp[8], Patrick Chaftari[9], Ray Hachem[1], Issam I Raad[1]

[1]Department of Infectious Diseases, Infection Control and Employee Health, The University of Texas MD Anderson Cancer Center, Houston, United States; [2]Integrative Medicine Program, The University of Texas MD Anderson Cancer Center, Houston, United States; [3]Hospital Medicine, The University of Texas MD Anderson Cancer Center, Houston, United States; [4]Pharmacy Clinical Programs, The University of Texas MD Anderson Cancer Center, Houston, United States; [5]Emergency Medicine-Research, The University of Texas MD Anderson Cancer Center, Houston, United States; [6]Gastroenterology Research, The University of Texas MD Anderson Cancer Center, Houston, United States; [7]Breast Surgical Oncology, The University of Texas MD Anderson Cancer Center, Houston, United States; [8]Cancer Therapeutics, The University of Texas MD Anderson Cancer Center, Houston, United States; [9]Department of Emergency Medicine, The University of Texas MD Anderson Cancer Center, Houston, United States

\*For correspondence:
achaftari@mdanderson.org

Competing interest: The authors declare that no competing interests exist.

## Abstract

**Background:** An increasing number of observational studies have reported the persistence of symptoms following recovery from acute COVID-19 disease in non-cancer patients. The long-term consequences of COVID-19 are not fully understood particularly in the cancer patient population. The purpose of this study is to assess post-acute sequelae of SARS-CoV-2 infection (PASC) in cancer patients following acute COVID-19 recovery.

**Methods:** We identified cancer patients at MD Anderson Cancer Center who were diagnosed with COVID-19 disease between March 1, 2020, and September 1, 2020, and followed them till May 2021. To assess PASC, we collected patients reported outcomes through questionnaires that were sent to patients daily for 14 days after COVID-19 diagnosis then weekly for 3 months, and then monthly thereafter. We also reviewed patients' electronic medical records to capture the persistence or emergence of new COVID19-related symptoms reported during any clinic or hospital encounter beyond 30 days of the acute illness and up to 14 months.

**Results:** We included 312 cancer patients with a median age of 57 years (18–86). The majority of patients had solid tumors (75%). Of the 312 patients, 188 (60%) reported long COVID-19 symptoms with a median duration of 7 months and up to 14 months after COVID-19 diagnosis. The most common symptoms reported included fatigue (82%), sleep disturbances (78%), myalgias (67%), and gastrointestinal symptoms (61%), followed by headache, altered smell or taste, dyspnea (47%), and cough (46%). A higher number of females reported a persistence of symptoms compared to males (63% vs. 37%; p=0.036). Cancer type, neutropenia, lymphocytopenia, and hospital admission during

acute COVID-19 disease were comparable in both groups. Among the 188 patients with PASC, only 16 (8.5%) were re-admitted for COVID-related reasons.

**Conclusions:** More than one out of two cancer patients, and more likely females, report PASC that may persist beyond 6 months and even 1 year. The most common symptoms are non-respiratory and consist of fatigue, sleep disturbance, myalgia, and gastrointestinal symptoms. Most of the cancer patients with PASC were managed on outpatient basis with only 8.5% requiring a COVID-19-related re-admission.

**Funding:** This research is supported by the National Institutes of Health/National Cancer Institute under award number P30CA016672, which supports the MD Anderson Cancer Center Clinical Trials Office. The funders had no role in study design, data collection, and interpretation, or the decision to submit the work for publication.

## Editor's evaluation

The authors provide valuable results from an observational study of 312 cancer patients to assess post-acute sequelae of SARS-CoV-2 infection (PASC). Although primarily descriptive and based on a relatively small sample, the study represents a solid expansion of the existing knowledge on persistent symptoms among cancer patients. The work will be of interest to oncologists working on long COVID.

## Introduction

Clinical outcomes of COVID-19 in patients with cancer remains an area of active study with several studies showing more severe outcomes and associated risk factors such as pre-existing comorbidities, cancer stage and therapy, and immunocompromised state, compared to the non-cancer population (*Dai et al., 2020*; *Petrilli et al., 2020*; *Wu and McGoogan, 2020*; *Zhou et al., 2020*).

The earliest studies characterized the acute and sub-acute effects of COVID-19 on multiple organ systems (*Gupta et al., 2020*) while more recent work has raised concern on the chronic symptoms attributed to COVID-19 that persist beyond the expected recovery period (*Barman et al., 2020*; *Nalbandian et al., 2021*) with some symptoms similar to those experienced during the recovery of other viral illnesses (*Xiong et al., 2021*; *Goërtz et al., 2020*; *Ahmed et al., 2020*; *Hui et al., 2005*; *Kausler et al., 1986*; *Moldofsky and Patcai, 2011*).

The long-term consequences of COVID-19 remain to be fully understood and there is no unequivocal consensus on the definition of post-acute sequelae of SARS-CoV-2 infection (PASC), with one adopted definition being the persistence of symptoms or new delayed complications that develop at least 4 weeks after a COVID-19 diagnosis (*Nalbandian et al., 2021*; *Datta et al., 2020*; *Greenhalgh et al., 2020*). The reported prevalence of PASC widely varies from 10% up to 87% in the general population (*Nalbandian et al., 2021*; *Xiong et al., 2021*; *Goërtz et al., 2020*; *Bowles et al., 2021*; *Carfi et al., 2020*; *Halpin et al., 2021*) and there is limited data available on PASC in cancer patients and how it affects their cancer progression, care, and treatment.

Several studies suggest the persistence of symptoms past 30 days in patients with severe initial COVID-19 symptoms and those that were hospitalized due to COVID-19 (*Barman et al., 2020*; *Carfi et al., 2020*; *Halpin et al., 2021*), while other studies have shown a prevalence of PASC among COVID-19 outpatients as well (*Xiong et al., 2021*; *Goërtz et al., 2020*; *Tenforde et al., 2020b*). Since cancer patients fall in a higher COVID-19 risk group (*Dai et al., 2020*) and in order to provide a better understanding of post-COVID-19 management among cancer patients, we sought to characterize the patterns of long COVID-19 in the specific cancer patient population.

## Methods

We identified patients with cancer receiving care at the University of Texas MD Anderson Cancer Center who were also diagnosed with COVID-19 disease between March 1, 2020, and September 1, 2020. None of the patients were vaccinated during the period of initial COVID diagnosis. We followed these patients as longitudinal cohort through patient-reported outcomes (PRO)-based remote symptom monitoring along with usual care with clinician visits. Patients were followed from March 2020 until

May 31, 2021. Patient questionnaires were sent out remotely daily for 14 days after COVID-19 diagnosis then weekly for 3 months, and then monthly thereafter. Chart reviews were conducted for each patient's encounter that included visits to our acute cancer care center, hospital re-admission, or clinic visit. Re-admissions were classified as either related or non-related to COVID-19 based on the reason for hospitalization as well as the reported signs and symptoms. PASC or long COVID-19 was defined as the persistence of COVID-19-related symptoms beyond 30 days of diagnosis or the emergence of new COVID19-related symptoms reported during a hospital or clinic encounter throughout the follow-up period that extended up to 14 months. COVID-19-related symptoms included, among others, fatigue, cough, chest tightness, dyspnea, headache, fever, altered smell or taste, myalgias, gastrointestinal symptoms (such as nausea, vomiting, or diarrhea), sleep disturbance, and limitations with activities of daily living. This study was approved by the institutional review board at MD Anderson. A waiver of informed consent was obtained since this study posed no risk to the patients given that no clinical or laboratory interventions were done.

### Statistical analysis

Categorical variables were compared using chi-square or Fisher's exact test, as appropriate. Continuous variables were compared using Wilcoxon rank sum test. All the tests were two-sided with a significance level of 0.05. The statistical analyses were performed using SAS version 9.4 (SAS Institute Inc, Cary, NC, USA).

## Results

Among the 602 patients who were diagnosed with COVID-19 during the study period, longitudinal data was collected on 312 patients that included 188 patients who developed PASC having reported symptoms that persisted at least 30 days after COVID-19 diagnosis and 124 who did not. The remaining patients (290) could not be followed or assessed beyond 30 days. The female gender rate was significantly higher in the PASC group compared to the non-PASC group (63% vs. 51%; p=0.036). The age and race were similar in both groups. The median age was 57, and 27% of both groups were above 65 years of age. While the rate of hypertension was higher in the non-PASC group (56% vs. 37%; p<0.001), the rate of other comorbidities such as COPD and congestive heart failures were similar in both groups. Furthermore, the type of underlying cancer was similar in both groups with the majority of patients having solid tumors. The type of underlying malignancy (hematological vs. solid tumor) as well as those with metastatic diseases were similar in both PASC and non-PASC groups. However, patients with PASC were significantly less likely to have refractory or relapse diseases (19% vs. 29%) In addition, the rate of neutropenia, lymphocytopenia, lower respiratory tract infections, hypoxia, oxygen requirement, inflammatory biomarkers, COVID-19-related hospital admissions, multiorgan failure as well as medical management of COVID-19 were similar in both groups (*Table 1*). Despite that, the rates of delay in cancer treatment after COVID were similar in both PASC and non-PASC patients (43% vs. 38%; p=0.44). Patients without PASC were more likely to have more acute severe COVID at diagnosis than patients who developed PASC (26% vs. 14%; p=0.009) and had a higher mortality beyond 30 days of initial COVID diagnosis (27% vs. 10%; p<0.0001). The 312 patients who were assessed beyond 30 days were followed for a median duration of 7 months and up to 14 months.

The most commonly reported symptoms among the 188 patients who developed PASC consisted of fatigue (82%), sleep disturbance (78%), myalgias (67%), gastrointestinal symptoms (62%), headache (47%), altered smell and taste (47%), dyspnea (47%), and cough (46%) (*Figure 1*).

Among the 188 cancer patients who reported PASC, 59 patients (31%) were re-admitted to the hospital during the follow-up period and beyond 30 days of acute illness, only 16 (8.5%) of the patients with PASC had a COVID-related re-admission.

## Discussion

Our data showed that 60% of 312 cancer patients diagnosed with COVID and followed up for a median duration of 7 month (up to 14 month) developed PASC. Females were more likely to report persistence of symptoms compared to males. However, cancer type, age, neutropenia, lymphocytopenia, hypoxia, severe disease, multiorgan failure, various interventions, or hospital admission during acute COVID-19 disease were not associated with higher risk of long COVID in our study group, while

**Table 1.** Comparing COVID-19 cancer patients with and without long-term symptoms (at least 30 days after their COVID-19 diagnosis).

| Variables | Having long-term symptoms | | p-Value |
|---|---|---|---|
| | No | Yes | |
| | (n=124) | (n=188) | |
| | N (%) | N (%) | |
| Age (years), median (range) | 57 (18–85) | 57 (21–86) | 0.47 |
| Age ≥65 | 33 (27) | 50 (27) | >0.99 |
| Gender | | | 0.036 |
| Male | 61 (49) | 70 (37) | |
| Female | 63 (51) | 118 (63) | |
| Race | | | 0.45 |
| Caucasian | 66 (53) | 102 (54) | |
| Black | 24 (19) | 26 (14) | |
| Hispanic | 28 (23) | 52 (28) | |
| Asian | 5 (4) | 8 (4) | |
| Other | 1 (1) | 0 (0) | |
| Prior COPD/bronchiolitis obliterans | 4 (3) | 7 (4) | >0.99 |
| History of hypertension | 70 (56) | 69/187 (37) | <0.001 |
| History of heart failure | 3 (2) | 5/186 (3) | >0.99 |
| Type of cancer | | | 0.92 |
| Hematological malignancy | 31 (25) | 48 (26) | |
| Solid tumor | 93 (75) | 140 (74) | |
| Metastasis | 42/93 (45) | 74/140 (53) | 0.25 |
| Status of cancer | | | 0.023 |
| Being treated initially | 65/123 (53) | 96 (51) | |
| Remission | 22/123 (18) | 56 (30) | |
| Refractory or relapse | 36/123 (29) | 36 (19) | |
| Chemotherapy within 1 year prior to COVID-19 | 74 (60) | 94 (50) | 0.09 |
| Immunotherapy within 6 months prior to COVID-19 | 7/123 (6) | 9/186 (5) | 0.74 |
| Radiation therapy within 6 months prior to COVID-19 | 12 (10) | 20 (11) | 0.78 |
| Severe COVID-19 at diagnosis | 32 (26) | 25 (14) | 0.009 |
| Cancer treatment delayed | 46/108 (43) | 66/174 (38) | 0.44 |
| Mortality* | 33 (27) | 18 (10) | <0.0001 |
| Hypoxia at diagnosis | 25 (20) | 23/180 (13) | 0.08 |
| Oxygen flow at diagnosis | | | 0.20 |
| High flow | 6/30 (20) | 6/43 (14) | |
| Low flow | 15/30 (50) | 15/43 (35) | |
| None | 9/30 (30) | 22/43 (51) | |
| Non-invasive ventilation at diagnosis | 16 (13) | 13/179 (7) | 0.10 |
| Lab values at COVID-19 diagnosis | | | |

*Table 1 continued on next page*

*Table 1 continued*

| Variables | Having long-term symptoms | | p-Value |
|---|---|---|---|
| ALC <1 K/µL | 44/74 (59) | 57/94 (61) | 0.88 |
| ANC <0.5 K/µL | 5/75 (7) | 5/89 (6) | >0.99 |
| Hemoglobin <10 g/dL | 23/65 (35) | 16/78 (21) | 0.06 |
| LDH ≥250 U/L | 29/58 (50) | 41/67 (61) | 0.21 |
| D-dimer ≥1 mcg/mL | 28/54 (52) | 27/61 (44) | 0.42 |
| Ferritin ≥500 ng/mL | 30/54 (56) | 36/59 (61) | 0.56 |
| CRP ≥40 mg/L | 32/56 (57) | 43/63 (68) | 0.21 |
| IL-6 ≥25 pg/mL | 18/41 (44) | 20/48 (42) | 0.83 |
| LRTI at diagnosis or progression to LRTI | 29 (23) | 35/182 (19) | 0.38 |
| Hospital admission | 52 (42) | 65 (35) | 0.19 |
| Remdesivir treatment | 17 (14) | 18 (10) | 0.26 |
| IL-6 pathway inhibitors – tocilizumab | 8 (6) | 5 (3) | 0.10 |
| Convalescent plasma | 8 (6) | 15 (8) | 0.61 |
| Steroids | 18 (15) | 19 (10) | 0.24 |
| Multiorgan failure | 6/121 (5) | 8/180 (4) | 0.84 |

*1. Patients who died within 30 days after COVID-19 diagnosis were excluded from this study. 2. Mortality status was determined within 14 months after COVID-19 diagnosis.

hypertension was more significantly associated with no PASC during follow-up. Furthermore, high-risk patients with baseline relapsing and refractory malignancy were associated with more acute severe COVID and higher mortality but less long-term COVID. The most common PASC symptoms were fatigue, sleep disturbances, myalgias, and gastrointestinal symptoms. Furthermore, among PASC patients who were re-hospitalized for any cause during the follow-up, the admission was COVID-related in only 27%.

In a manner similar to our current study, the CDC reported that 60% of patients with immunosuppressive conditions continue to have COVID-19 symptoms beyond the acute illness (*Tenforde et al., 2020b*). In the same CDC study, hypertension was found to be significantly less often associated with PASC which is another similar finding noted in our study. It is unknown from a pathophysiologic perspective as to why hypertension, which is a risk factor for acute and severe COVID-19, should be less associated with PASC. However given the fact that the pathogenesis of acute COVID-19 is related to the interaction of the viral spike protein with the angiotensin-converting enzyme 2 (ACE 2), one could postulate that this pathway does not play an important role in long COVID compared to the residual inflammatory processes that seem to continue to occur in PASC (*Nalbandian et al., 2021*; *Kaseb et al., 2021*). Similarly, males are at greater risk of acute complications and death associated with acute COVID-19 possibly related to higher ACE 2 expression levels in the male gender (*Kaseb et al., 2021*; *Jin et al., 2020*; *Viveiros et al., 2021*). However, in our study females were more likely to develop PASC also suggesting a different mechanism of action for PASC besides ACE 2 expression (*Nalbandian et al., 2021*).

The existing literature has also shown a higher prevalence of PASC and a longer time to recovery among those with more severe COVID-19 who required hospital admission and medical interventions (*Barman et al., 2020*; *Carfi et al., 2020*; *Halpin et al., 2021*). However, there has been several reports of PASC among patients with self-reported COVID-19 who had mild acute illness and were never hospitalized (*Xiong et al., 2021*; *Goërtz et al., 2020*; *Tenforde et al., 2020b*; *Tenforde et al., 2020a*). In our study, severe acute COVID associated with hypoxia, multiorgan failure, hospital admission, or underlying risk factors for the progression of acute COVID-19 in cancer patients such as neutropenia, lymphocytopenia, hematologic malignancy, or older age were not associated with increased risk of long COVID.

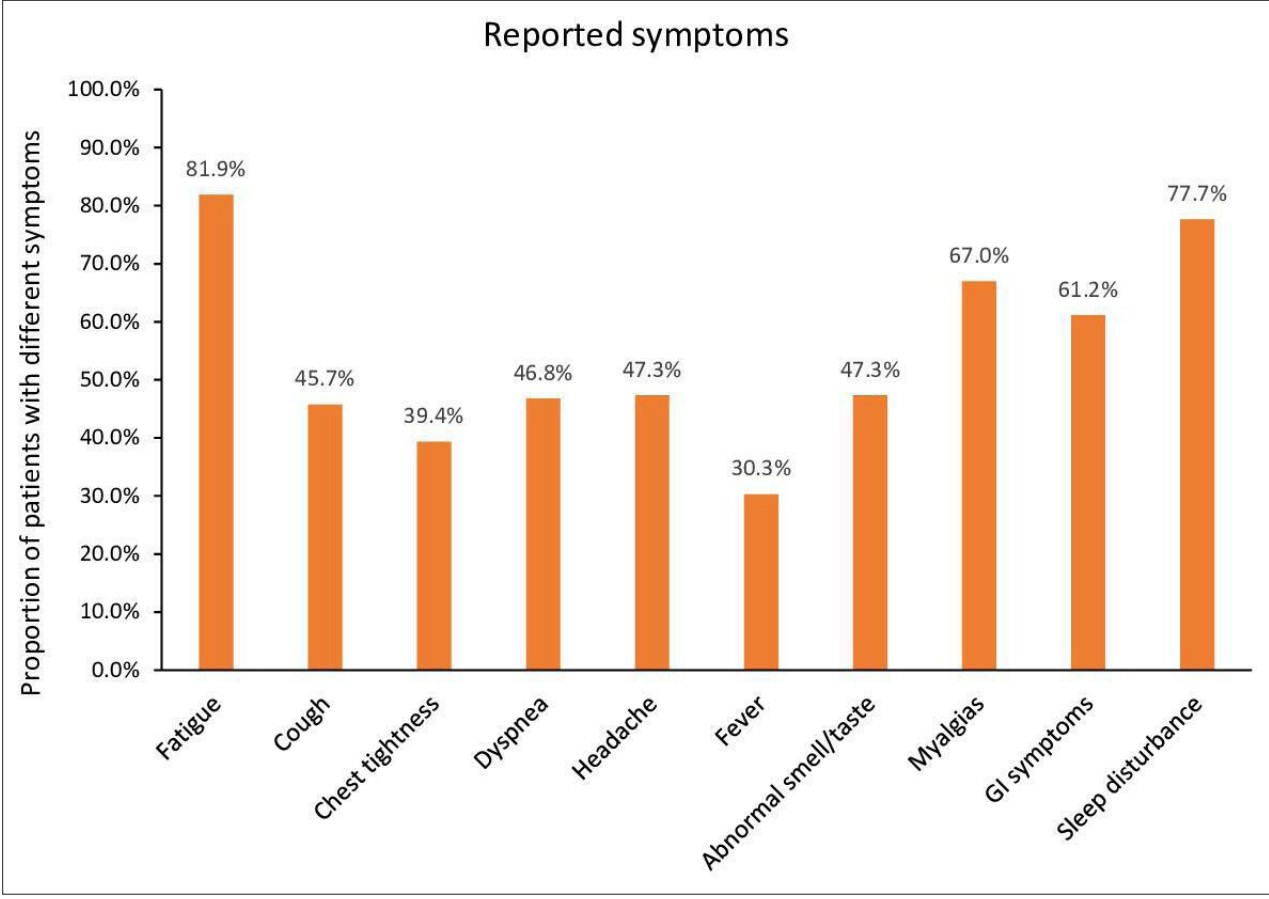

**Figure 1.** Post-acute sequela of SARS-CoV-2 infection (PASC)-related symptoms.

In the general population and following acute COVID-19, PASC symptoms occurred at a variable rate ranging from 10% to 87% (*Nalbandian et al., 2021*; *Xiong et al., 2021*; *Goërtz et al., 2020*; *Bowles et al., 2021*; *Carfi et al., 2020*; *Halpin et al., 2021*). In a manner similar to our results, fatigue was the most prevalent PASC symptom in many reports occurring in up to 87% of COVID-19 patients compared to 82% in our patient population (*Nalbandian et al., 2021*; *Goërtz et al., 2020*; *Carfi et al., 2020*; *Halpin et al., 2021*). Other common PASC symptoms reported in our study such as sleep disturbances, myalgias, gastrointestinal symptoms, headache, loss of smell and taste, dyspnea, and cough were also widely and commonly reported in the COVID-19 literature (*Nalbandian et al., 2021*; *Xiong et al., 2021*; *Goërtz et al., 2020*; *Bowles et al., 2021*; *Carfi et al., 2020*; *Halpin et al., 2021*). However, all of these symptoms commonly occur in patients with underlying malignancy who continue to receive treatment with conventional chemotherapy, radiotherapy, and immunotherapy (including checkpoint inhibitors). Hence, the rates of some of these PASC symptoms in the cancer patient population could be exaggerated and partially related to the underlying malignancy and its treatment.

In our cancer patient population, PASC was not associated with a high rate of COVID 19-related hospital admissions during the follow-up. All cause admissions occurred at a rate of 31% among our cancer patients with PASC and among those who were admitted only 27% were admitted because of COVID-19-related reasons. Hence, among all PASC patients in our study only 8.5% were admitted for COVID 19-related reasons. Furthermore, some of those few that were admitted for COVID-19-related reasons could have had reinfection with a new variant during the prolonged follow-up period or reactivation related to their immunosuppressed status. Hence, even in our high-risk cancer patient population, most of the symptoms associated with PASC were managed for the most part on an outpatient basis without the need for hospital admission.

Our study has its limitations. First, the retrospective nature of this study limits any firm conclusions about factors directly contributing to PASC in our COVID-19 cancer patients. Second, this is a single

center study which could also limit the generalization of our results. Third, the subjective nature of surveys in the absence of quality of life assessment scales and other verified objective scales to assess reported PASC criteria could also limit our conclusions. Finally, the presence of several confounding causes of hospital admissions and overlapping symptoms in cancer patients that may be due to their underlying disease, aggressive treatments, and comorbidities present their own challenge in identifying PASC among cancer patients post their COVID-19 diagnosis.

In conclusion, long COVID occurred in the majority of our cancer patients diagnosed with acute COVID-19 with a preponderance of symptoms (such as fatigue, sleep disturbances, and gastrointestinal symptoms) over a long time period. Long COVID was less likely in patients who experienced severe acute COVID at diagnosis and who had a higher mortality rate. Most of the patients with long COVID/PASC were fairly and adequately managed on outpatient basis without the need for hospital admission. Besides the female gender, we found no other underlying condition or severity of illness during acute COVID-19 that would predict PASC.

## Acknowledgements

We thank Ms Salli Saxton, Department of Infectious Diseases, Infection Control and Employee Health, MD Anderson Cancer Center, Houston, for helping with the submission of the manuscript. This research was supported by the National Institutes of Health/National Cancer Institute under award number P30CA016672, which supports MD Anderson Cancer Center's Clinical Trials Office.

## Additional information

### Funding

| Funder | Grant reference number | Author |
|---|---|---|
| National Institutes of Health | P30CA016672 | Hiba Dagher<br>Anne-Marie Chaftari<br>Ishwaria M Subbiah<br>Alexandre E Malek<br>Ying Jiang<br>Peter Lamie<br>Bruno Granwehr<br>Teny John<br>Eduardo Yepez<br>Jovan Borjan<br>Cielito Reyes-Gibby<br>Mary Flores<br>Fareed Khawaja<br>Mala Pande<br>Noman Ali<br>Raniv Rojo<br>Daniel D Karp<br>Ray Hachem<br>Issam I Raad |

The funders had no role in study design, data collection and interpretation, or the decision to submit the work for publication.

### Author contributions

Hiba Dagher, Conceptualization, Writing - original draft, Writing - review and editing, critical review, commentary acquisition of data; Anne-Marie Chaftari, Conceptualization, Writing - review and editing, oversight and leadership responsibilities, including mentorship interpretation of data; Ishwaria M Subbiah, Conceptualization, Writing - original draft, Writing - review and editing, interpretation of data; Alexandre E Malek, Conceptualization, Writing - original draft, Writing - review and editing, critical review, commentary acquisition of data; Ying Jiang, Data curation, Software, Formal analysis, Visualization, Writing - original draft, Writing - review and editing; Peter Lamie, Visualization, Writing - review and editing, critical review, commentary acquisition of data; Bruno Granwehr, Writing - review and editing, critical review, commentary acquisition of data; Teny John, Writing - review and editing, critical review, commentary acquisition of data; Eduardo Yepez, Writing - review and editing, critical

review, commentary acquisition of data; Jovan Borjan, Writing - review and editing, critical review, commentary acquisition of data; Cielito Reyes-Gibby, Writing - review and editing, critical review, commentary acquisition of data; Mary Flores, Writing - review and editing, critical review, commentary acquisition of data; Fareed Khawaja, Writing - review and editing, critical review, commentary acquisition of data; Mala Pande, Writing - review and editing, critical review, commentary acquisition of data; Noman Ali, critical review, commentary acquisition of data; Raniv Rojo, Writing - review and editing, critical review, commentary acquisition of data; Daniel D Karp, Writing - review and editing, critical review, commentary; Patrick Chaftari, Data curation, Formal analysis; Ray Hachem, Conceptualization, Writing - original draft, Writing - review and editing, interpretation of data critical review, commentary oversight and leadership responsibilities, including mentorship; Issam I Raad, Conceptualization, Writing - original draft, Writing - review and editing, Interpretation of Data oversight and leadership responsibilities, including mentorship

### Author ORCIDs
Anne-Marie Chaftari http://orcid.org/0000-0001-8097-8452
Cielito Reyes-Gibby http://orcid.org/0000-0003-4500-6476

### Ethics
Human subjects: This study was approved by the institutional review board at MD Anderson. A waiver of informed consent was obtained since this study posed no risk to the patients given that no clinical or laboratory interventions were done.

### Decision letter and Author response
Decision letter https://doi.org/10.7554/eLife.81182.sa1
Author response https://doi.org/10.7554/eLife.81182.sa2

---

## Additional files

### Supplementary files
• MDAR checklist

### Data availability
Data Availability Statement: The study protocol, statistical analysis plan, lists of deidentified individual data, generated tables and figures will be made available upon request by qualified scientific and medical researchers for legitimate research purposes. Requests should be sent to https://faculty.mdanderson.org/profiles/anne-marie_chaftari.html and yijiang@mdanderson.org. Data will be available on request for 6 months from the date of publication. Investigators are invited to submit study proposal requests detailing research questions and hypotheses in order to receive access to these data.

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
