## [Editor Report]

The authors provide valuable results from an observational study of 312 cancer patients to assess post-acute sequelae of SARS-CoV-2 infection (PASC). Although primarily descriptive and based on a relatively small sample, the study represents a solid expansion of the existing knowledge on persistent symptoms among cancer patients. The work will be of interest to oncologists working on long COVID.

---

## [Decision Letter]

**Decision letter after peer review:**

Thank you for submitting your article "Long COVID in Cancer patients: Preponderance of Symptoms in Majority of Patients over Long Time Period" for consideration by *eLife*. The evaluation has been overseen by a Reviewing Editor and a Senior Editor. The following individual involved in review of your submission has agreed to reveal their identity: Nawale Hajjaji.

As is customary in *eLife*, the reviewers have discussed their critiques with one another. What follows below is the Reviewing Editor's edited compilation of the essential and ancillary points provided by reviewers in their critiques and in their interaction post-review. Please submit a revised version that addresses these concerns directly. Although we expect that you will address these comments in your response letter, we also need to see the corresponding revision clearly marked in the text of the manuscript. Some of the reviewers' comments may seem to be simple queries or challenges that do not prompt revisions to the text. Please keep in mind, however, that readers may have the same perspective as the reviewers. Therefore, it is essential that you attempt to amend or expand the text to clarify the narrative accordingly.

Essential revisions:

Overall this was a good manuscript that could be improved by kindly addressing the concerns of the reviewer below.

The following comments could be helpful to strengthen the work:

1) Whether cancer stage or the type of anticancer therapy are associated to PASC is of great interest; these data should be within reach and added to Table 1.

2) Some patients might have participated in a vaccination trial with mRNA vaccine between March and September 2020. If this is the case, this information should be added to Table 1.

3) As mentioned by the authors in the introduction section line 94 "there is limited data available on PASC in cancer patients and how it affects their cancer progression, care and treatment". The addition of these analyses would greatly strengthen the article.

4) It could be interesting to distinguish severe and mild covid at diagnosis to compare their pattern of long covid. In fact, the frequency of long covid reported in cancer patients with mild covid was only 22% (doi: 10.3389/fonc.2022.901426), much less than the frequency reported in this article.

---

## [Author Response]

The following comments could be helpful to strengthen the work:1) Whether cancer stage or the type of anticancer therapy are associated to PASC is of great interest; these data should be within reach and added to Table 1.

The type of underlying malignancy (hematological vs solid tumor) as well as those with metastatic diseases were similar in both PASC and non-PASC groups. However, patients with PASC were significantly less likely to have refractory or relapse diseases (19% vs 29%) Added to the Results section. High risk patients with baseline relapsing and refractory malignancy tend to be associated with more acute severe COVID and higher mortality but less long term COVID. Added to the Discussion section.

2) Some patients might have participated in a vaccination trial with mRNA vaccine between March and September 2020. If this is the case, this information should be added to Table 1.

None of our patients were vaccinated between March and September 2020 (added in methods).

3) As mentioned by the authors in the introduction section line 94 "there is limited data available on PASC in cancer patients and how it affects their cancer progression, care and treatment". The addition of these analyses would greatly strengthen the article.

We thank the reviewer for their comments. We extracted additional data and found that the rates of delay in cancer treatment after COVID were similar in both PASC and non-PASC patients (43% vs 38%; p=0.44). Mortality beyond 30 days of initial COVID diagnosis was significantly higher in patients without PASC compared to those with PASC (27% vs 10%; p<0.0001). Added in the results and Discussion section We have added the name of an author (Patrick Chaftari) who helped collect the additional data.

4) It could be interesting to distinguish severe and mild covid at diagnosis to compare their pattern of long covid. In fact, the frequency of long covid reported in cancer patients with mild covid was only 22% (doi: 10.3389/fonc.2022.901426), much less than the frequency reported in this article.

We thank the reviewers for their interesting comments. Patients without long COVID were more likely to have more acute severe COVID at diagnosis than patients who developed long COVID (26% vs 14%; p=0.009). Added in results and Discussion section.